# Study of Elevation Forces and Resilience of the Schneiderian Membrane Using a New Balloon Device in Maxillary Sinus Elevations on Pig Head Cadavers

Erick Rafael Fernández Castellano [1,*], Magaly Teresa Marquez Sanchez [2] and Javier Flores Fraile [1]

1   Department of Surgery, University of Salamanca, 37007 Salamanca, Spain; j.flores@usal.es
2   Salamanca Biomedical Research Institute, 37007 Salamanca, Spain; maglymarquez77@gmail.com
*   Correspondence: er.fernandez@usal.es

**Abstract: Background**: Although elevation of the sinus can be considered a predictable procedure, it is nonetheless not free of complications, for which reason there is a constant search for new tools and techniques that may reduce these complications. The present study focused on maxillary sinus lifts performed on pig heads cadavers, using a new device with the balloon technique. **Materials and Methods**: Fifteen ex vivo adult pig heads were used in this experimental study. Sinus floor elevation was performed using the new balloon elevation control system, which consists of a syringe containing latex and serum as well as a system of burs for membrane access and control. Each lift was performed within a 3 min time frame while constant pressure was applied to allow the tissue to adapt to the tension. **Results**: In 100% of cases, perforations do not occur during aperture or in the elevation of the wall. In the global sample, there was histological elevation in 73.33% compared to 26.66% non-elevation ($p = 0.0268$). **Conclusions**: Within the limits of this study, the maxillary sinus lifts employing the new device and the balloon technique were minimally invasive procedures. The elevations achieved proved sufficient to allow future placement of implants of varying lengths and diameters without risk of perforating the membranes, even in the presence of crests of less than 1 mm.

**Keywords:** balloon elevation; sinus floor elevation; schneiderian membrane; pigs head; ex vivo

## 1. Introduction

Currently, rehabilitation with dental implants is an effective therapeutic option for the replacement of missing teeth in patients with partial or total edentulism. For this to be achieved, the success and survival of these implants must be ensured, with one crucial factor being sufficient bone availability to bear functional loads [1–3]. The loading forces which occur during chewing have been shown to require implants of sufficient length, within 8–10 mm for the premolar and molar regions in both the mandible and the maxilla, these being generally accepted as standard measurements [4,5].

Given that patients frequently present with bone defects of variable characteristics as a result of various processes such as tooth loss, which leads to an average 40–60% decrease in horizontal and vertical bone loss of the alveolar crest during the first year, periodontal disease, trauma, tumors, and cysts, the correct three-dimensional placement of implants often becomes complicated [6–10]. Because of this, various surgical techniques have been proposed, such as onlay–inlay bone grafts, distraction osteogenesis, inferior alveolar nerve transposition, split crest, guided bone regeneration (GBR), and maxillary sinus augmentation [11–15].

This last procedure, which is one of the most used in implant practice, arises from the need to rehabilitate the posterior maxillary areas which are often atrophic due to the prevalence of type IV cancellous bone and to dental extractions which cause rapid vertical and horizontal resorption, in many cases accompanied by increased pneumatization of the sinus in these areas [12–16]. This procedure was first described by Boyne et al. in 1965,

becoming popular in the 1980s. In it, the Caldwell-Luc procedure was used to access the sinus. Summers, for his part, popularized the crestal approach together with the use of osteotomes and bone grafts, a procedure which is considered less invasive [16–18].

Although the sinus lift procedure can be considered clinically predictable, it is not exempt from complications such as the dispersion of graft material in the sinus cavity, wound dehiscence, hematomas, migration of implants inside the sinus, fenestrations, oroantral fistulae, epistaxis, bone sequestration, acute sinusitis, and perforations of the Scheiderian membrane [19–22]. This last complication is considered the main drawback of this procedure, occurring in 10–35% of cases where the direct method is used, with an approach through the lateral wall of the maxillary sinus [23,24]; or in 25% of cases with an indirect approach through the alveolar crest and where the initial bone height lay between 3 and 6 mm [25–28]. Elevation of the mucosa of the maxillary sinus inevitably causes stretching of the tissue which, in cases of higher elevations, may easily lead to tearing and perforation of the sinus membrane [29]. A retrospective study reported similar findings: the highest rates of perforation were correlated to higher elevations. These perforations often lead to contamination of graft tissue and to the subsequent loss of both the graft and the implants [30,31].

The present study based on maxillary sinus lifts was performed on ex vivo bisected pig heads, employing a new device for the balloon technique patented by the author. The study was justified by the need to assess the efficacy of this new system, which is proposed as both a more conservative technique for the elevation of the Schneiderian membrane, and a novel procedure which currently has no articles proving its reliability. The study aims to assess the feasibility of the device, describe the technique, and contrast its advantages and disadvantages.

## 2. Materials and Methods

An experimental study was carried out by a single operator. Fifteen ex vivo heads of adult pigs were used. The pig is well established in implant research as an animal model. The maxillary sinus of adult pigs is known to provide a volume of up to 30 cm$^3$ sufficient for the elevation procedure and a soft tissue lining comparable to that of humans. The pig heads were obtained from the local butchery immediately after slaughter. Hence, consultation with the appropriate ethics committee was not necessary. The bisected heads were sold as a base material for food processing by the slaughterhouse.

Due to the pronounced height of the alveolar crest in adult pigs, an approach to the maxillary sinus through the lateral wall of the sinus was preferred for the experiments. The lateral approach to the sinus allowed a greater control over the dissection of the membrane.

All animals were free of local and systemic diseases to prevent any bias due to pathological alterations of the tissue. The experiments were carried out within a standard 6 h post-mortem under a constant room temperature of 21 °C.

Sinus floor elevation was performed with the new elevation control system patented by the author and registered in the Spanish Official Industrial Property Gazette (patent application: 202130571). This system consists of a syringe containing stops, retaining elements, latex, and serum (see Figure 1a).

To access the sinus, a Bien-Air turbine with a 3.5 mm diameter diamond bur was used to outline the window, with abundant irrigation, followed by a donut-type bur with 20:1 reduction NSK contra-angle at 2000 rpm, 35 Ncm torque with abundant irrigation, and a 3 mm stop to prevent rupture of the membrane. The sinus was accessed at 2 cm perpendicular to the gingival margin of the maxillary first molar (see Figure 1b).

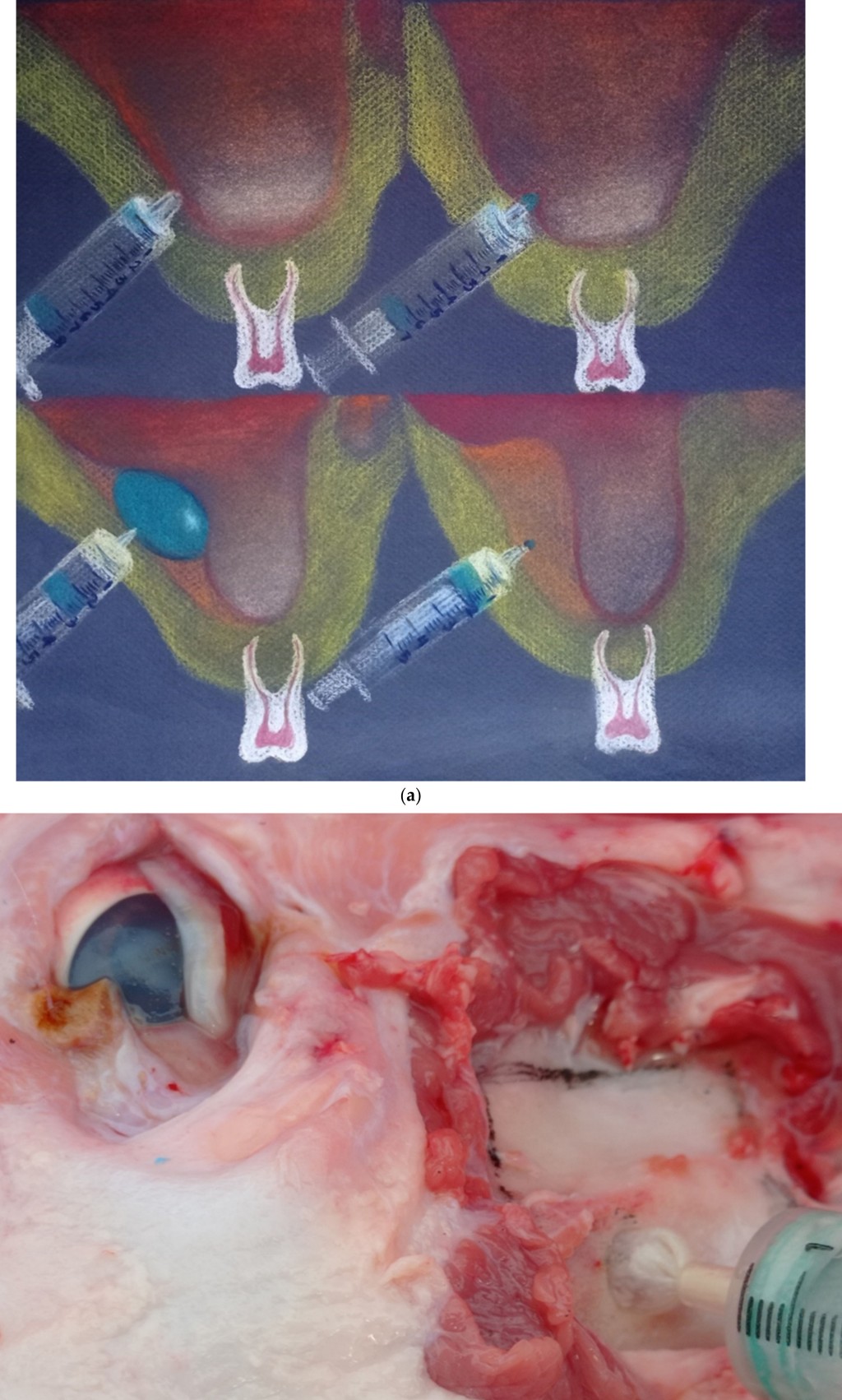

(a)

(b)

**Figure 1.** *Cont.*

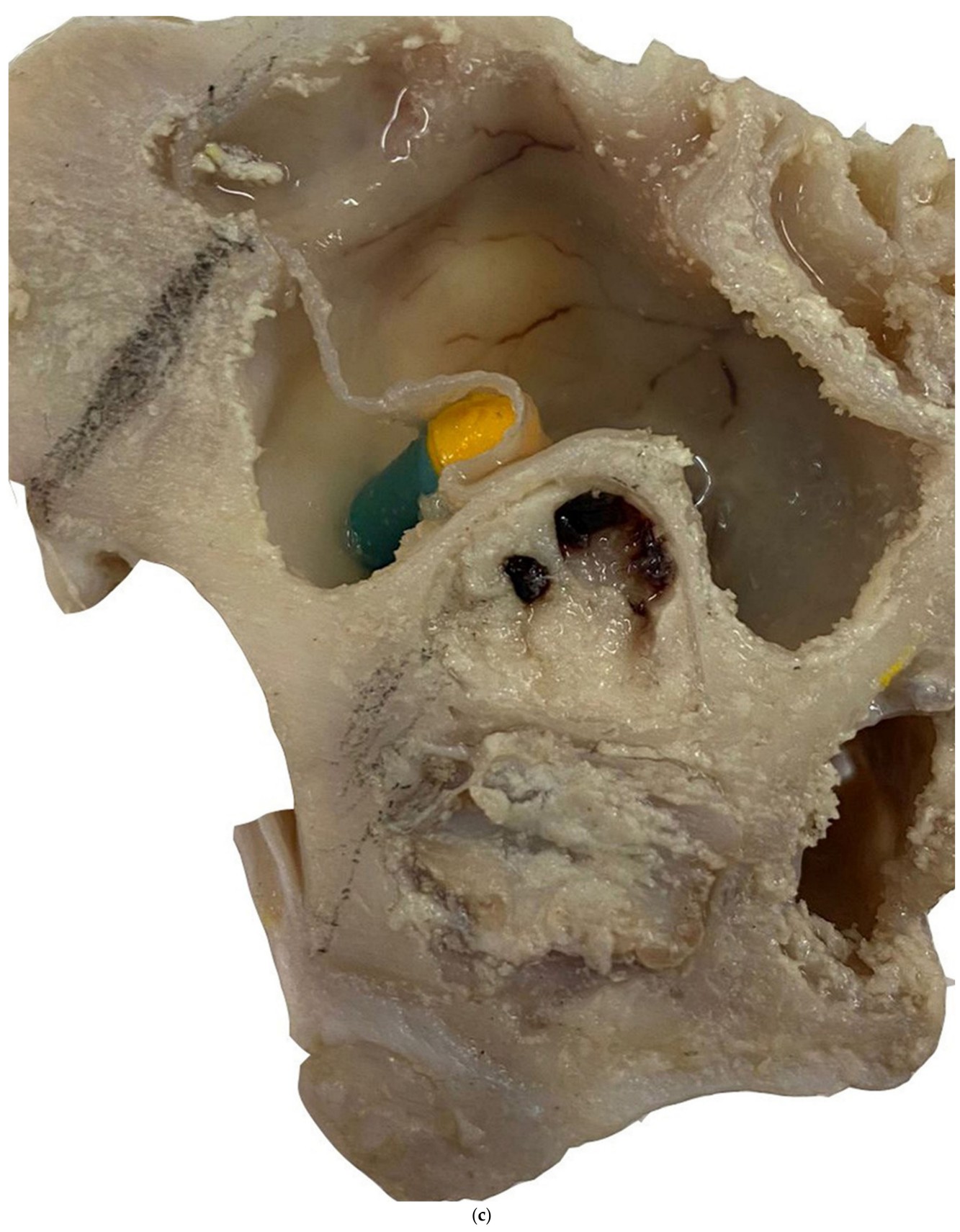

(**c**)

**Figure 1.** *Cont.*

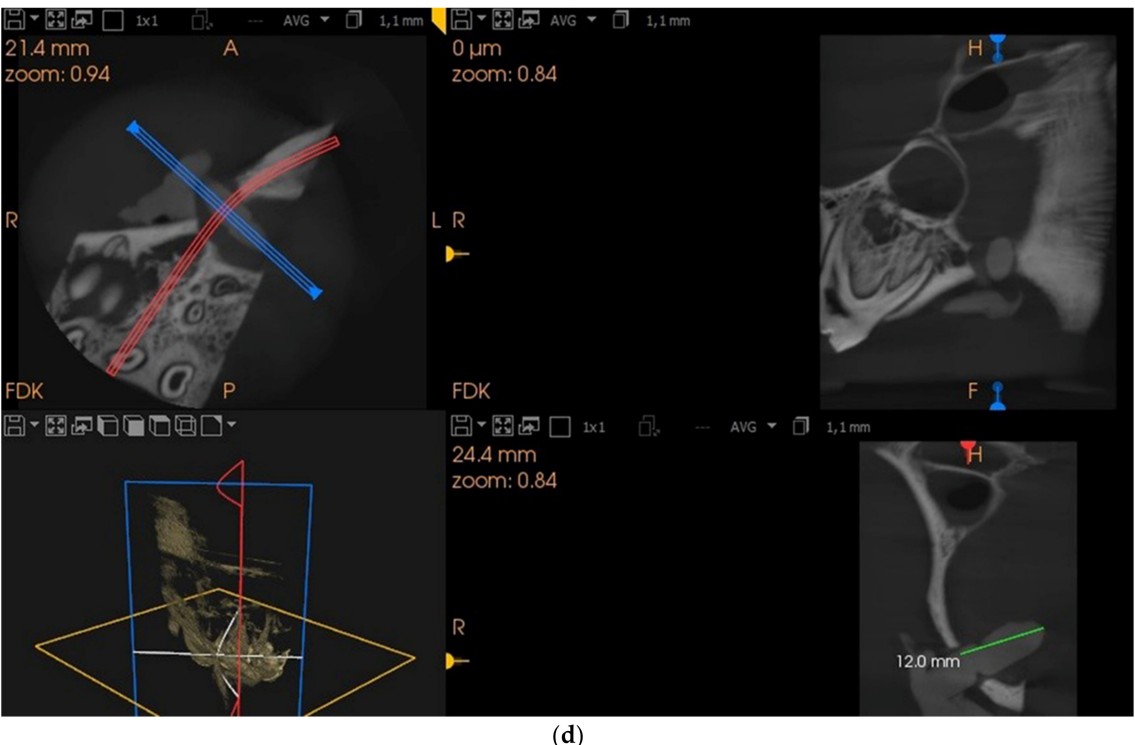

(**d**)

**Figure 1.** (**a**) New lifting device invented by the author. From left to right. Insertion of the device into the sinus, beginning of inflation, maximum pressure and inflation, deflation of the balloon after elevation. (**b**) Elevation window. (**c**) Section for histological analysis. (**d**) CBCT scans, 5 × 5. Oblique cut.

### 2.1. Measurement of Balloon Pressure and Resistance

A manometer was connected to a lateral outlet valve on the syringe, allowing maximum measurement of pressure throughout the experiment, with the aim of recording pressure changes during the separation of the sinus mucosa from the sinus floor. The pressure monitoring equipment used for this study consisted of an AZ 82152 digital pressure gauge (Resolution: Bar: 1.034 (0.001), Range: 0~15 psi, Accuracy: 0.3% FS at 25 C, Linearity/Hysteresis: 0.29~1%FS, Repeatability 0.2~0.5%FS, Combined Accuracy 1.0% FS).

The zero point of the pressure gauge was set to ambient pressure during each experiment to rule out any influence. Following insertion into the maxillary sinus as described above, the balloon was progressively filled to a target volume of 2 mL of serum, while pressure was applied. The pressure exerted during balloon filling and the resulting elevation of the mucosa were constantly monitored and recorded. Each elevation was carried out within a 3 min timeframe while constant pressure was applied to allow the tissue to adapt to the altered tension and separate from the surface of the bone. Once this time had elapsed and the elevation procedure was complete, the balloon and the syringe were removed from the sinus.

To prevent bias with regard to pressure data obtained during monitoring of the elevation, a pilot trial was required to measure the influence of wear on the balloon material, and the resistance of this material to the recorded pressure. To this end, the balloon was filled three times consecutively with 2 mL of serum and the resulting maximum pressures were recorded: 318.2, 315.3, and 320.7 mmHg, respectively, with a mean of 318.1 mmHG. This preliminary experiment was performed in ex vivo pig heads with no tissue or other material around the balloon. The acquired pressure data were interpreted as the elastic resistance of the balloon material to volumetric expansion. The pressure recorded in this pilot experiment was subtracted from the maximum pressure measured during the ex vivo elevation experiment in order to determine the force originating solely from the mucosa.

The elevation heights were obtained by carefully stretching the mucosa and introducing a standard impression material (Elite HD + Putty soft regular set, Zhermack, Rovigo, Italy) in

the space below the mucosa. This material was subsequently dried and cut with a jigsaw into pieces measuring approximately $4 \times 4 \times 4$ cm for histological study, with a safety margin sufficient to avoid accidental damage to the mucosa (see Figure 1c).

The $5 \times 5$ CBCT scans were then taken with a CS 8100 3D unit in order to measure the elevations obtained in the samples (see Figure 1d).

### 2.2. Histological Processing: Non-Decalcified Bone Samples Embedded in Plastic

The bone samples obtained following the slaughter of the experimental animals were preserved in formalin for fixation and conservation.

Since these bone samples were non-decalcified, they were washed in water and sawn to macroscopically delimit the study areas. The samples were then dehydrated by immersion in ethanol solutions of increasing concentration (70°, 80°, 90°, and 95°), soaking for 24 h in each solution until reaching absolute ethanol, where they remained for 2 days.

The samples were then placed for 15 days under agitation in liquid polymethyl metacrilate (PMMA) at 4 °C. After 15 days of immersion, the bone samples were transferred to glass cylinders with a base of solid PMMA and filled with liquid PMMA. The tubes were sealed with parafilm and kept in an oven at 32 °C for 5–6 until polymerization. The polymerized block with the bone sample was then cut with a microtome (Microm HM 350 S). The first sections were made with a thickness of 30 μm until reaching the study area, where 5 μm sections were cut and placed on gelatinized slides, covered with polyethylene film and pressed and dried at 60 °C for 24 h.

For histological staining, the sections were deplasticized by immersion in methyl acetate for 55 min, then washed in decreasing alcohol solutions until distilled water. The stain used was Goldner's Trichrome, which allows osteoid to be distinguished from calcified bone, as well as the study of the morphology and distribution of cells in the tissues.

Microscopic images of histological samples were captured using a Nikon Digital Sight DS-smc camera coupled to a Nikon Eclipse 90i optical microscope.

### 3. Results

Fifteen $5 \times 5$ scans were obtained with a CS 8100 3D unit, measuring the thickness of the sinus wall and the elevation height which was obtained (see Table 1). The mean thickness of the sinus walls studied was 1.45 mm $\pm$ 0.53, median 1.20 mm, range 0.90–2.40 mm, while the mean elevation achieved was 10.6 mm. The mean height was 10.63 mm $\pm$ 1.32, median 10.20, range 8.00–13.30 (see Tables 1 and 2 and Figure 2a).

**Table 1.** Study of sinus wall thickness and heights achieved measured with CBCT.

| Sinus | Sinus Wall Thickness (mm) | Elevation Height Measured with CBCT (mm) | Membrane Perforated during Aperture | Membrane Perforated during Elevation | Elevation of All Layers. Histological Analysis |
|---|---|---|---|---|---|
| 1 | 1 | 10.6 | No | No | Yes |
| 2 | 1.1 | 11 | No | No | No |
| 3 | 1 | 10.1 | No | No | Yes |
| 4 | 0.9 | 10.7 | No | No | Yes |
| 5 | 1.8 | 10.1 | No | No | Yes |
| 6 | 0.9 | 12.5 | No | No | No |
| 7 | 2.2 | 9.7 | No | No | Yes |
| 8 | 2.4 | 8 | No | No | Yes |
| 9 | 1.2 | 13.3 | No | No | Yes |

**Table 1.** *Cont.*

| Sinus | Sinus Wall Thickness (mm) | Elevation Height Measured with CBCT (mm) | Membrane Perforated during Aperture | Membrane Perforated during Elevation | Elevation of All Layers. Histological Analysis |
|---|---|---|---|---|---|
| 10 | 1.8 | 9.8 | No | No | No |
| 11 | 1.1 | 10.1 | No | No | No |
| 12 | 1 | 11.8 | No | No | Yes |
| 13 | 1.4 | 10.2 | No | No | Yes |
| 14 | 1.7 | 12 | No | No | Yes |
| 15 | 2.3 | 9.6 | No | No | Yes |

**Table 2.** Shows a comparison between the thickness of the sinus floor ($p$ = 0.000002) and the elevation height measured with CBCT ($p$ = 0.0014). There was a significant difference of 0.97 mm between sinus floor thickness groups. There were significant differences of 2.08 mm in the elevation height.

| | Sinus Floor Thickness (mm) | | Elevation Height Measured with CBCT (mm) | |
|---|---|---|---|---|
| **Men** | 1.45 | | 10.63 | |
| SD | 0.53 | | 1.32 | |
| Median | 1.20 | | 10.20 | |
| Minimum value | 0.90 | | 8.00 | |
| Maximum value | 2.40 | | 13.30 | |
| Groups | **Thickness < 1.45 mm** | **Thickness > 1.45 mm** | **Height < 10.63 mm** | **Height > 10.63 mm** |
| Mean | 1.06 | 2.03 | 9.80 | 11.88 |
| SD | 0.15 | 0.30 | 0.73 | 0.95 |
| Median | 1.00 | 2.00 | 10.10 | 11.90 |
| Minimum value | 0.90 | 1.70 | 8.00 | 10.70 |
| Maximum value | 1.40 | 2.40 | 10.60 | 13.30 |
| *p*-value T-student | 0.000002 | | 0.0014 | |

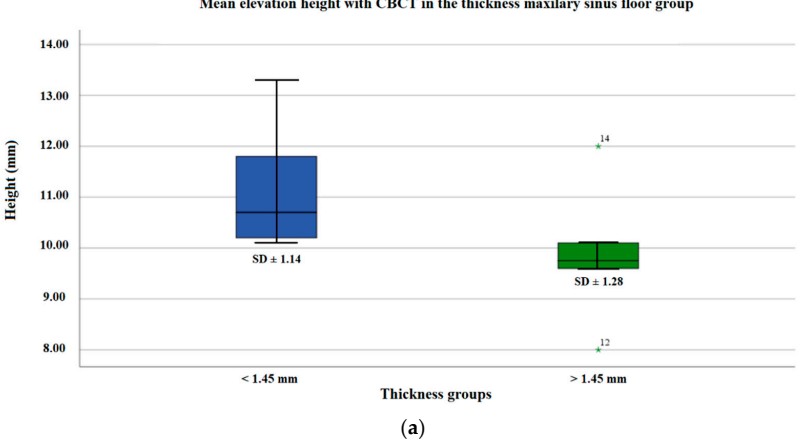

(**a**)

**Figure 2.** *Cont.*

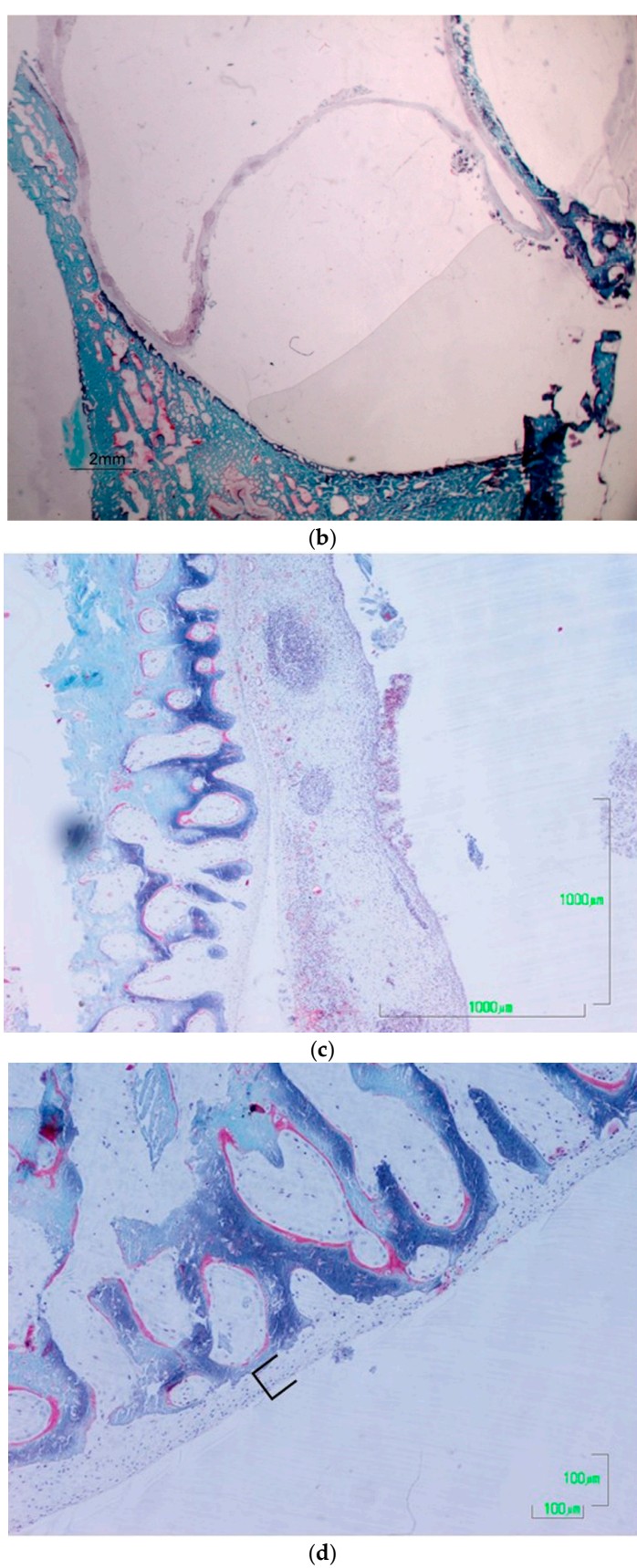

**Figure 2.** (**a**) Distribution of the mean elevation height with the CBCT (mm) in the maxillary sinus floor thickness groups. (**b**) Osteotomy, elevated mucosa 100 μm. (**c**) Elevation start zone 100 μm. (**d**) Periosteum 100 μm.

In 100% of cases, perforations do not occur during aperture or in the elevation of the wall. There was histological elevation in all layers in 53.33% of the >10.63 mm height group compared to 46.67% of the <10.63 mm height group ($p$ = 1.0000); there was no elevation in 86.67% of the >10.63 mm height group compared to 13.33% in the <10.63 mm height group. In the global sample, there was histological elevation in 73.33% compared to 26.66% non-elevation ($p$ = 0.0268) (see Table 3, Figure 3a,b).

**Table 3.** Distribution and comparison of membrane perforation and histology within groups for maxillary membrane height and global sample.

| Global Sample | | | |
|---|---|---|---|
| **Variables** | ***n*** | **%** | ***p*-Value** |
| Membrane perforated during aperture (no) | 15 | 100.00 | 1.000 |
| Membrane perforated during elevation (no) | 15 | 100.00 | |
| Elevation of all layers. Histological analysis (no) | 4 | 26.66 | 0.0268 |
| Elevation of all layers. Histological analysis (yes) | 11 | 73.33 | |

| **Subgroups** | **Height < 10.63 mm** | | **Height > 10.63 mm** | | **Total** | | **Fisher's Exact Test** |
|---|---|---|---|---|---|---|---|
| **Variables** | ***n*** | **%** | ***n*** | **%** | **%** | ***n*** | ***p*-Value** |
| Membrane perforated during aperture (no) | 9 | 60.00 | 6 | 40.00 | 15 | 100.00 | 0.4661 |
| Membrane perforated during elevation (no) | 9 | 60.00 | 6 | 40.00 | 15 | 100.00 | 0.4661 |
| Elevation of all layers. Histological analysis (no) | 2 | 13.33 | 13 | 86.67 | 15 | 100.00 | 0.0001 |
| Elevation of all layers. Histological analysis (yes) | 7 | 46.67 | 8 | 53.33 | 15 | 100.00 | 1.0000 |

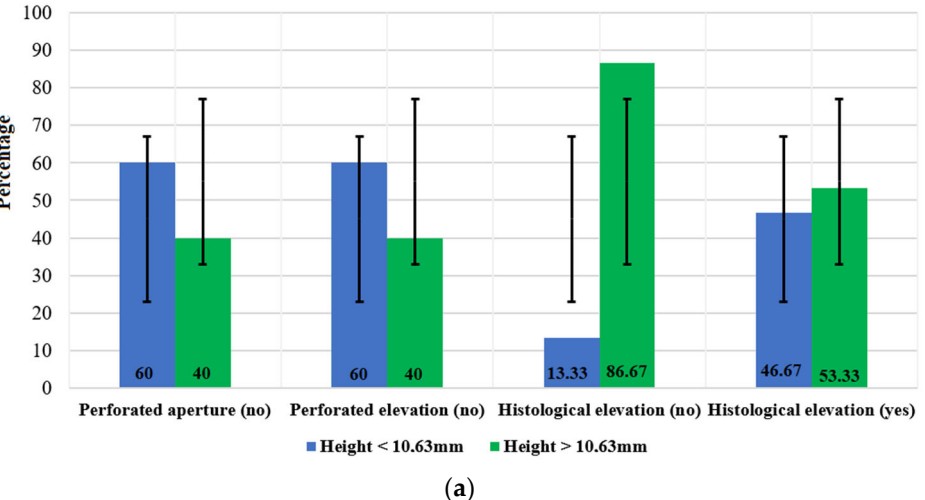

(**a**)

**Figure 3.** *Cont*.

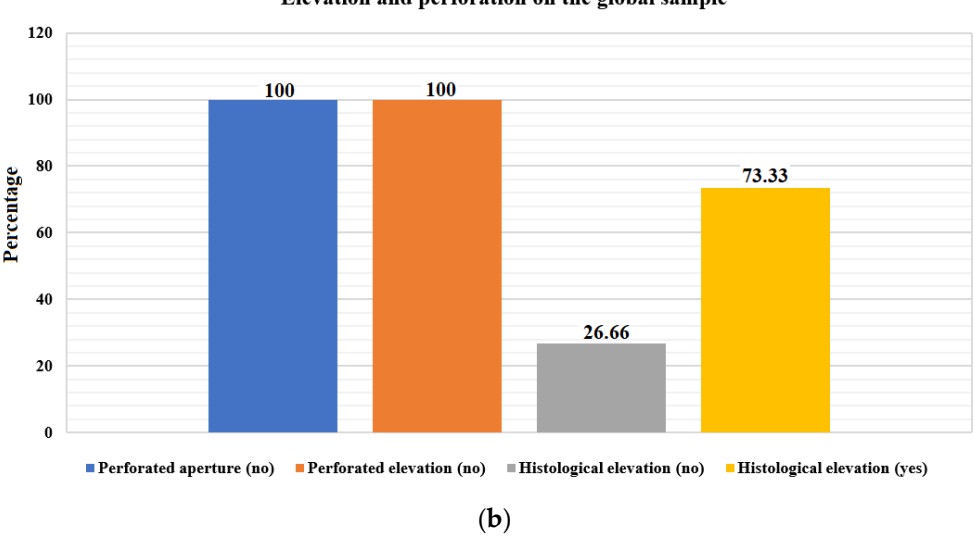

**(b)**

**Figure 3.** (**a**) Distribution of the histological elevation and the membrane perforated in the height groups, (**b**) Distribution of the histological elevation and the membrane perforated on the global sample.

Microscopic images of the histological samples did not reveal ruptures or laceration of the membranes in any of the samples. Only in four cases was it impossible to elevate all the layers, with the glandular layer left unelevated and a clearer view of the epithelium in the elevated mucous membranes. A thin layer of collagen tissue was observed in the bone, which may be interpreted as the periosteum or parts of it (see Figure 2b–d).

The mean maximum pressure obtained during the balloon experiments was 662.62 mmHg. The maximum values did not exceed 726.20 mmHg. The minimum monitored pressure was 598.80 mmHg. The pressure required to overcome the resistance of the balloon material, as determined during the pilot test, was subtracted from the maximum pressure. Therefore, the mean pressure required to elevate the maxillary bone floor mucosa was calculated to be 344.46 mmHg in this ex vivo animal model. No mucosal tearing was observed. Pressure values over time and fill volume showed a strong initial rise in pressure. In 5 of the 15 cases, there was an Underwood septum in the region where the mucosa was elevated. However, the elevation process in regions with Underwood septa (mean pressure = 368.12 mmHg) did not produce significantly different elevation forces ($p$ = 0.096) compared to sinus mucosal elevations without the presence of septa (mean pressure = 332.63 mmHg) (see Tables 4 and 5, Figure 4a–c)

**Table 4.** Pressure values in mmHg.

|  | **Max** | **Septum** | **Pressure on Membrane** |
|---|---|---|---|
| SINUS 1 | 598.8 | NO | 280.7 |
| SINUS 2 | 620.2 | NO | 302.1 |
| SINUS 3 | 680.2 | SI | 362.1 |
| SINUS 4 | 650.7 | NO | 332.6 |
| SINUS 5 | 601.9 | NO | 283.8 |
| SINUS 6 | 630.8 | NO | 312.7 |
| SINUS 7 | 654.9 | SI | 335.8 |
| SINUS 8 | 635.4 | NO | 317.3 |
| SINUS 9 | 726.2 | SI | 408.1 |
| SISUS 10 | 659.7 | NO | 341.6 |

**Table 4.** *Cont.*

|  | Max | Septum | Pressure on Membrane |
|---|---|---|---|
| SINUS11 | 704.4 | NO | 386.3 |
| SINUS 12 | 658.2 | SI | 340.1 |
| SINUS 13 | 706.8 | NO | 388.7 |
| SINUS 14 | 698.6 | NO | 380.5 |
| SINUS 15 | 712.6 | SI | 394.5 |

**Table 5.** Distribution and comparison of maximum pressures and pressures on the membrane in the global sample and in the sinuses with and without a septum.

|  | Maximum Pressure mmHg | | Pressure on Membrane | |
|---|---|---|---|---|
| **Mean** | 662.62 | | 344.46 | |
| SD | 40.86 | | 40.88 | |
| Median | 658.20 | | 340.10 | |
| Minimum value | 598.80 | | 280.70 | |
| Maximum value | 726.20 | | 408.10 | |
| Groups | Without septum | With septum | Without septum | With septum |
| Media | 650.73 | 686.42 | 332.63 | 368.12 |
| SD | 50.88 | 32.00 | 40.88 | 32.24 |
| Median | 643.050 | 680.20 | 324.95 | 362.10 |
| Minimum value | 598.80 | 654.90 | 280.70 | 335.80 |
| Maximum value | 706.80 | 726.20 | 388.70 | 408.10 |
| *p*-value | 0.093 | | 0.096 | |

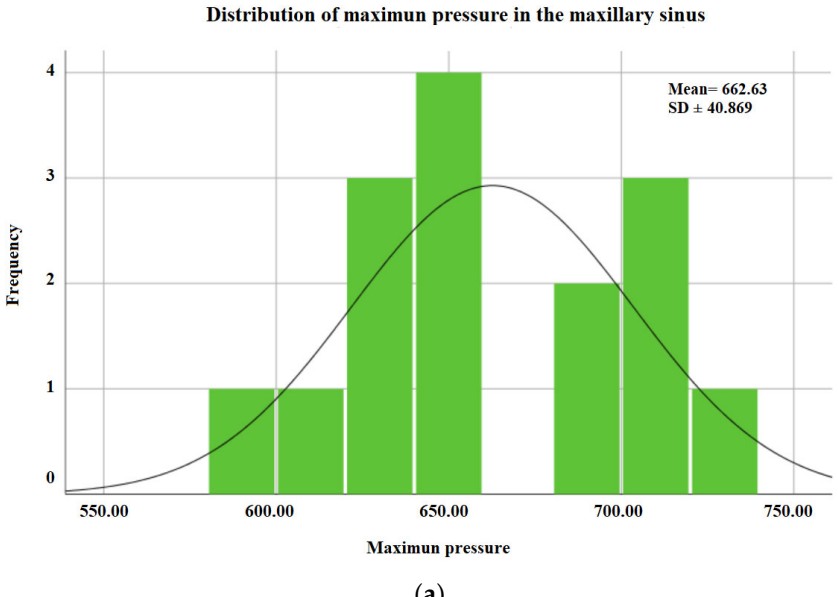

(a)

**Figure 4.** *Cont.*

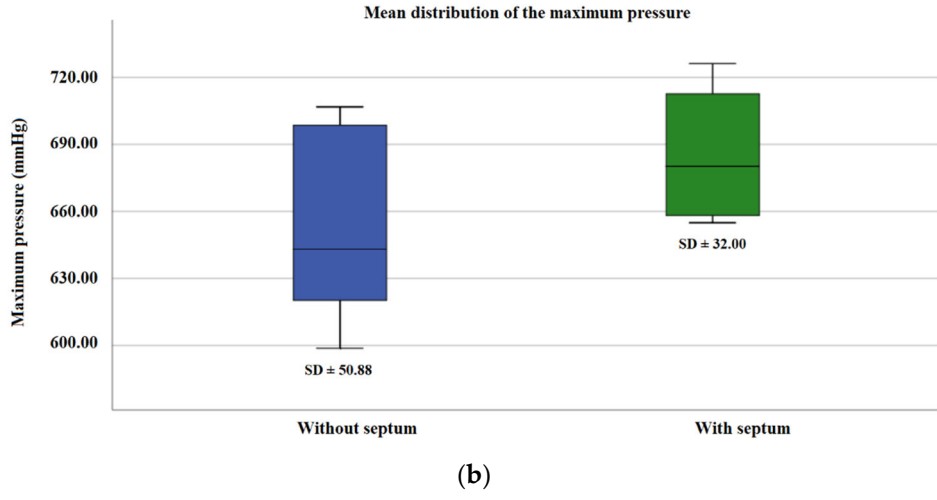

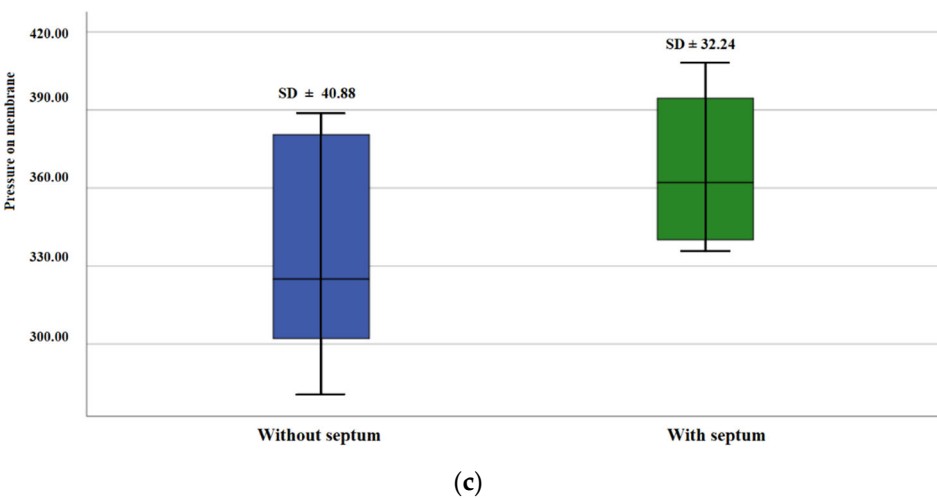

**Figure 4.** (**a**) General distribution of maximum pressure in the maxillary sinus. (**b**) Distribution of mean maximum pressure in the maxillary sinus with and without a septum. (**c**) Mean pressure distribution in the maxillary sinus membrane with and without a septum.

## 4. Discussion

With a mean elevation height of 10.6 mm, no membrane lacerations could be found during gross and microscopic examinations. The mean pressure during maxillary sinus augmentation was therefore assumed to be below the soft tissue tear point for this animal model. However, the results of the balloon experiment should be interpreted with caution, as they were obtained from an ex vivo animal model and changes in tissue resistance and stability due to post-mortem degradation and dehydration of proteins cannot be ruled out. Pig heads are a well-established animal model in implantology research due to their similarity with human anatomy [32–34]. Nevertheless, the mucosa of the maxillary sinus of the pig is known to be thicker than that of humans [35,36]. This can lead to increased adhesion and elastic tensile forces.

In pigs, there is a greater number of Underwood septa. However, the Underwood septa in the region of the lifting procedure did not show a significant influence ($p = 0.096$) on the lifting process or the lifting force required in this study. Therefore, it is assumed that this anatomical variation may have no influence on sinus elevation procedures where a balloon system is used. Similar results were noted by Stelzle et al. in their study of pressure and resilience with the Meisinger system [31].

Due to the pronounced crestal bone height found in this animal model, the lateral surgical approach through the maxillary sinus wall was chosen. In this area, the bone

level in the pig would not exceed 2–3 mm, reflecting the situation found in a patient with an atrophied maxillary alveolar process [36,37]. All other steps of the procedure were performed and assessed analogous to a crestal approach. However, the lateral accessed may have influenced the pressure values obtained, as the thickness of the lateral sinus mucosa may be different to that of the sinus floor. The thickness of healthy sinus membrane in humans has been shown to vary depending on the region of the maxillary sinus which is being studied (0.99 to 2.58 mm), as well as between individuals [38,39].

The mean pressure needed to elevate the mucosa of the maxillary bone floor was calculated to be 344.5 mmHg, a considerably lower value than that obtained in the pressure and resilience measurement study by Stelzle et al., where the balloon elevation system (Meisinger) was used on ex vivo bisected pig heads, giving a pressure of 748 mmHg [31].

Histological images did not reveal ruptures or lacerations of the membranes in any of the samples. Thin layers of collagen were observed in the bone, which may be interpreted as periosteum or parts of it, in agreement with the study by Stelzle et al. which compared various methods of indirect elevation in an ex vivo experimental study.

Despite the above limitations, the preliminary results of this study with an experimental set-up based on the author's patented sinus floor elevation kit with an inflatable balloon demonstrate that the measuring configuration is appropriate for directly and reliably determining the pressures which occurred in the sinus during the elevation procedure. The possible transfer of this procedure to a clinical environment has therefore been proven, along with a demonstration of the general applicability of this technical configuration for future clinical trials.

## 5. Conclusions

Within the limits of this study on ex vivo bisected pig heads, maxillary sinus elevations using the new device and balloon technique were found to be minimally invasive procedures, with no membrane perforations observed in the histological analyses, even in the presence of a septum and of residual crests less than 1 mm. The heights achieved, analyzed histologically and measured with CBCT, proved sufficient to allow the future placement of implants of varying lengths and diameters.

## 6. Patents

Sinus floor elevation was performed with the new elevation control system patented by the author and registered in the Spanish Official Industrial Property Gazette (patent application: 202130571). This system consists of a syringe containing stops, retaining elements, latex, and serum.

**Author Contributions:** Conceptualization, E.R.F.C.; literature search, E.R.F.C.; data analysis, M.T.M.S.; writing—original draft preparation, E.R.F.C.; writing—review and editing, E.R.F.C. and J.F.F. All authors have read and agreed to the published version of the manuscript.

**Funding:** This research received no external funding.

**Institutional Review Board Statement:** Not applicable.

**Informed Consent Statement:** Not applicable.

**Data Availability Statement:** Not applicable.

**Conflicts of Interest:** The authors declare no conflict of interest.

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
