# Peer review of "Study of Elevation Forces and Resilience of the Schneiderian Membrane Using a New Balloon Device in Maxillary Sinus Elevations on Pig Head Cadavers"

_applsci, doi:10.3390/app12094406_

Round 1
Reviewer 1 Report
The Reviewer would like to thank the authors for their efforts in performing the study. During the reviewing process, some points have been raised that require correction or modification.
TITLE
ex vivo bisected pig heads: Better to use pig head cadavers.
ABSTRACT
The study was experimental, prospective, controlled, and single-centre: This is just an ex vivo experiment not prospective or controlled or single centre.
15 ex vivo adult pig heads were used: Fifteen as it's the beginning of a sentence (numbers at the beginning of sentences are written in letters).
Results. Histological elevation of all layers: What does that mean?
Keywords: They need to be changed. Please use also Schneiderian membrane, Pigs, cadaver, ex-vivo.
INTRODUCTION
to prevent dehiscences and fenestrations: Please replace this sentence with one that fits sinus elevation as what is missing here is the adequate amount of the bone for implant placement in cases of sinus elevation but not dehiscence or fenestration.
in horizontal and vertical layers of the alveolar crest: horizontal and vertical bone loss of alveolar crest.
traumas: Trauma
Please use the correct format for the references inserted in the text. Use square brackets instead of curl brackets and separate the references by comma.
maxillary sinus augmentation. (1-3,12-15): These are too many references. Please use only one reference for each technique.
The present study centred on: This is not scientific.
MATERIALS AND METHODS
An experimental, prospective, controlled, single-centre study was carried out by a single operator: Please check the comment in the abstract.
A favourable opinion from the university Research Ethics
Committee was not required for the study, as the heads were purchased from the local butcher’s shop as a base ingredient for food processing: The paragraph is not written in a scientific way.
All animals were free of local and systemic diseases to prevent any bias due to pathological alterations of the tissue: How did you know that the pigs were free from any pathological condition related to the sinus.
room temperature of 21ºC: Is not the normal room temperature is 25?
This preliminary experiment was performed in vitro: In-vitro or ex-vivo:
Histological Assessment of the membrane
Ive to admit that part of the experiments does not totally fit here. I would accept the radiographic assessment which is fine but how could you do histological assessment for a cadaver. This is a sacrificed animal which means it's dead and talking about inflammatory cells here is non-sense. Please remove this whole experiment from the study.
Please use normal graphs not tilted one and add standard deviation to the bars.
Author Contributions: The main author designed and patented the device: Please use the authors guidelines to write down the author contribution.
Author Response
Thank you very much for the appreciation. Everything that is suggested was modified. Best regards.

Reviewer 2 Report
The article presents itself as a well-studied topic. Although the idea of ​​using a balloon is not new, both the study and the patenting of the system are to be appreciated. The great advantage of the system is that it minimizes the risk of perforation of the sinus membrane. Elevation of the sinus mucosa is and will remain a topical issue.
Author Response
THANK YOU VERY MUCH FOR YOUR APPRECIATION
Reviewer 3 Report
A very well designed study and written manuscript. However, despite the claim in the write up, the technique is not new and have been previously described for example by Smiler and Kfir. You may need to refer to those studies and compare your device and explain its superiority.
Also, it is not clear wethere the device is able to elevate the membrane wide enough in all three dimensions for proper future implant placement or was only able to elevate it over the device as shown in the Figure 1.C making it an unfavorable elevation.
Author Response
Thank you very much for your appreciation.

Round 2
Reviewer 1 Report
The Reviewer would like to thank the authors for their efforts in revising the manuscript.